# The impact of green technology innovation on carbon emission reduction capacity in China: Based on spatial econometrics and threshold effect analysis

**Wei Wei[1], Yu Ma[2]***

**1** School of Accountancy, Henan University of Engineering, Zhengzhou, China, **2** School of Accountancy, Anyang Institute of Technology, Anyang, China

* ycon666@163.com

**Data Availability Statement:** All relevant data are within the manuscript.

**Funding:** The Key Scientific Research Project of Colleges and Universities in Henan Province

## Abstract

Dual carbon targets is the key to sustainable development in China and even the world, so how to achieve green low-carbon economy is a general trend. As an important guarantee for carbon emission reduction, green technology innovation is essential for promoting the development of green economy. The analysis of the internal mechanism of green technology innovation affecting carbon emission reduction will help provide the development direction for green technology innovation, help China's carbon neutrality and carbon peak goals as soon as possible, and contribute Chinese wisdom to the sustainable development of the world. This study innovatively adds the carbon market vitality in measuring carbon reduction capacity, and the study extends the previous carbon reduction capacity to expand the influence of green technology innovation and is more practical, and explores the relationship between green technology innovation and carbon reduction capacity. Based on 30 provinces in China from 2006 to 2021, SDM and panel threshold model are constructed to analyze the spatial spillover effect of green technology innovation and carbon emission reduction ability, the threshold characteristics of green technology innovation on carbon emission reduction ability. The results show the carbon emission reduction capacity of Chinese provinces and cities has been increasing year by year, showing the situation of "leading in eastern China, rising of central China, and catching up with western China". Green technology innovation and carbon emission reduction capacity show positive spatial correlation, which is manifested in the fact that green technology innovation not only reduces the carbon emission reduction capacity of the region effectively but also has significant spatial spillover characteristics on promoting the carbon emission reduction capacity of the neighboring regions. Also, green technology innovation have a double threshold effect on carbon emission reduction capacity, which means its influence on carbon emission reduction capacity is obviously strengthened. Based on such study conclusions, it is recommended to build a new pattern of "great synergy", strengthen the in-depth integration of green technology innovation and carbon emission reduction, and formulate green and low-carbon differentiated standards, so as to promote a comprehensive green transformation.

"Research on the influence and mechanism of scientific and technological innovation on the high-quality economic development in the Yellow River Basin" (No.:23A790032) received by Y. M. Soft Science Research Project of Henan Province "Study on the interactive spillover effect and improvement path of ecological protection and high-quality development in the YRB" (No.: 232400410004) received by Y. M. The funders had no role in study design, data collection and analysis, decision to publish, or preparation of the manuscript.

**Competing interests:** The authors have declared that no competing interests exist.

## 1. Introduction

Global warming is seriously threatening the homeland and future of human beings, which is irreversible and inextricably linked to carbon emissions caused by human activities [1]. Meanwhile, the global energy crisis has raised the uncertainty of carbon emission to a certain extent [2]. As the second largest economy, China had a total of 12.1 billion tons of carbon emissions in 2022 according to the data, which ranked first in the world and keeps growing. Carbon emissions have more harmful effects globally, such as being a major cause of extreme weather and grain reduction, and even affecting people's healthy lives [3].

In view of the seriousness of the global carbon situation, China put forward dual carbon policy in 2015, aiming at reducing carbon emissions, improving environmental quality, and realizing sustainable growth. In 2020, government renewed his commitment to achieving dual carbon goals, which not only conforms to China's urgent determination to pursue a green transformation but also reflects the responsibility of a great nation. Therefore, the enhancement of a country's ability to reduce carbon emissions has become a growing concern. This paper will focus on the following issues: how to evaluate the carbon emission reduction capacity, whether green technology innovation will produce adjacent carbon emission reduction effect, and whether the impact of green technology innovation on carbon emission reduction capacity is unchanged.

The construction of green technology innovation is like a raging fire, providing impetus for ecological and environmental management. Essentially, green technology innovation integrates environmental factors and green elements into technological innovation to promote low-carbon. Therefore, promoting the rapid improvement of carbon emission reduction capacity is an important basis for ecological civilization construction. Analyzing the impact of green technology innovation on carbon emission has important practical value for enhancing carbon emission reduction capacity.

## 2. Literature review

Carbon emissions are affected by a variety of factors, like urbanization development, energy structure, trade, and other factors [4–9], with innovation being one of the key factors [10].

According to the existing conclusions, some scholars believe that technology innovation is not limited to technology, but also an improvement to the organizational structure [11] that can significantly improve carbon emissions [12–16]. Therefore, the use of technology innovation to promote carbon reduction is an urgent problem at present [17]. Yu et al. constructed a strategy of emission reduction technological innovation and found through simulation design that technology innovation can boost carbon emission reduction [18]. Specifically speaking, technology innovation can reduce carbon emissions and even generate a snowball effect [19, 20], through the enhancement of carbonless, carbon reduction, and negative carbon technologies [21]. Some scholars have found that green innovation can decrease the carbon reduction cost by constructing a supply chain models [22].

Some scholars believe that technology innovation can also reduce carbon emissions by influencing other factors. Liu et al. applied a spatial econometric model and a mediator model and found that technology innovation not only contributes to carbon emission reduction in region but also has an impact on carbon emission reduction in around areas with heterogeneity [23]. Technology innovation can directly affect carbon emissions and promote carbon emission reduction by affecting investment [24]. In addition, improving energy efficiency can be used to reduce carbon. What's more, technology innovation improves regional and corporate carbon emissions, which also holds true in terms of long-term and short-term results [25, 26].

However, the exact relationship between them has not been unified conclusion [27]. Some scholars believe that technology innovation has the possibility to add or reduce carbon

emissions [28–30], and generate energy saving and emission reduction effect at a certain level of innovation. However, technology innovation will rise carbon emissions with the role of expanding scale of production [31]. Samargandi also believed technology innovation does not necessarily reduce carbon emissions [32], it is beneficial to carbon emissions in the long but not short term [33]. Sahoo used the OLS model and found that innovation can promote carbon emissions [34]. Research on the Relationship between them has been applied in agriculture and industry [35].

Carbon emissions and green innovation is positive correlation [36], green innovation technology including energy conservation and emission reduction, CO2 capture technology, is the key to repair regional ecological, reduce carbon emissions [37], can reduce carbon emissions [38], and the inhibition in China and Europe and other countries or regions have long-term stability [39, 40]. Green technology innovation can reduce carbon emissions [41] by improving energy efficiency. Some scholars have studied the heterogeneity of green innovation, and found that cooperative green innovation is more effective than independent green innovation [42]. In addition, carbon pollution shows significant agglomeration characteristics [43].

To summarize, there are still shortcomings although there is a wealth of research related to technology innovation and carbon emissions. Firstly, the existing studies mainly focus on the relationship between technology innovation and carbon emissions, but it is short of assay of the relationship of green technology innovation on carbon emission reduction capacity. Some scholars have ignored the fact that the meaning of carbon emission reduction has changed. Compared with carbon emission reduction, carbon emission reduction capacity is sustainable and can provide reference for the world. Secondly, the improvement of carbon emissions will be effected by surrounding neighboring provinces. However, the existing studies only take into account the direct effect of the region and tend to ignore the spatial spillover effect, lack of regional integration and even global integration development pattern thinking, resulting in biased results [44]. Finally, the traditional linear models, which were believed to address the link between technology innovation and carbon emissions by most scholars, actually have difficulty explaining the complex relationship between the two because of the ignorance of the cumulative nature of innovation and the effect of heterogeneity.

The marginal contributions of this study lie in three aspects. First, from the research perspective, this paper earlier research green technology innovation affect carbon reduction ability, green provide new application direction for green science and technology innovation, a few research focus on green technology innovation of carbon reduction effect, but focus on carbon emissions as a research object, has certain limitations, this paper on carbon emission reduction capacity, better reflect the green technology innovation on the ability of carbon emission reduction, more research value; Second, from the perspective of research methods, the spatial effect model and panel threshold model are used to conduct empirical analysis of 30 provinces in China to investigate the direct effect, spillover effect and threshold characteristics of green technology innovation on carbon emission reduction capacity; Third, from the perspective of research significance, it is of great significance to clarify the mechanism effect of the regional carbon emission reduction capacity, which is of great significance for promoting the comprehensive transformation of economic and social development and the harmonious coexistence between man and nature.

## 3. Theoretical mechanism and research hypothesis

### 3.1. Direct impact of green technology innovation on carbon emission reduction capacity

Green technology innovation includes technological upgrading and any innovation that can realize environmental protection or energy consumption saving [45]. Green technology

innovation has an effect on carbon emission reduction capacity by the reduction of carbon emission reduction cost and technology scale effect. To be specific, in the case of a certain cost of production of traditional petrochemical energy, the research development of new energy could reduce the cost of clean energy, thus lowering the green premium lower and improving carbon emission reduction capacity. Green technology innovation can also directly reduce carbon emissions by facilitating the transformation of industries and enhancing the in-depth integration of digitalization and green industrialization. In addition, the reduced industry's dependence on resources with green technology innovation and consumers' preference for green products make enterprises more motivated to implement green technology innovation. In short, green technology innovation accelerates decarbonization with technology as its core, thus addressing the carbon emission problem at a very low cost. From this, a hypothesis is formulated as follows:

H1: Green technology innovation has a direct carbon reduction impact on the region.

## 3.2. Spatial spillover impact of green technology innovation on carbon emission reduction capacity

Carbon reduction has a strong spatial dependence [46], the similarity of industrial structure in neighboring regions makes carbon emissions have certain spatial correlation characteristics. Under the background of regional integration development, inter-regional green technology innovation cooperation has become the norm. The integrated industrial chain of government, industry, academia, research, and utilization promotes the in-depth integration of the innovation chain and financial chain, thus accelerating the R&D efficiency of green technology innovation. Moreover, the positive externalities of carbon governance make environmental benefits, except for technological spillovers, an incentive for governments to strengthen regional cooperation. There is an "imitation effect" when governments compete and cooperate regionally, i.e., they introduce or innovate their technologies to catalyze the low-carbon industries. Therefore, hypothesis H2 is put forward as follows:

H2: Green technology innovation has a positive impact on carbon emission reduction in neighboring regions.

## 3.3. Non-linear impact of green technology innovation and carbon emission reduction capacity

Technology innovation is accumulation process, and so is green technology innovation, which call for the integration of innovation resources and elements to reduce innovation costs and improve the green R&D efficiency. With the deepening of green technology innovation, more talent and funds will be attracted to gather, which will increase the stock of innovation resources through the siphon effect [47]. Then, the increase in the stock of innovation resources also produces a scale effect, that is, the profit tendency of innovation resources makes them flow to more efficient regions, thus producing more green technology achievements [48].

Based on this, the hypothesis H3 is put forward as follows:

H3: The impact of green technology innovation on carbon emission reduction capacity has non-linear characteristics at different stages to be specific, there is a certain promotion effect when it is at a low level, and its promotion effect will be further enhanced when it exceeds a certain level.

## 4. Research design

### 4.1. Research methods

**4.1.1. Exploratory spatial data analysis.** Geary's C and Moran's I statistics were applied to perform spatial correlation test of carbon emission reduction and green technology innovation. Global Moran's I was calculated:

$$Moran's\ I = \frac{[n\sum_{i=1}^{n}\sum_{j=1}^{n}W_{ij}(x_i - \bar{x})(x_j - \bar{x})]}{[\sum_{i=1}^{n}\sum_{j=1}^{n}W_{ij}\sum_{i=1}^{n}(x_i - \bar{x})^2]} \tag{1}$$

Where n represents research institutes, W weigh the spatial matrix, $x_i$ and $x_j$ weighs the values of i and j variables, with $\bar{x}$ as the mean value. Moran's I statistic takes values in the range of $[-1, 1]$.

Geary's C was calculated by the Equation:

$$Geary's\ C = \frac{[(n-1)\sum_{i=1}^{n}\sum_{j=1}^{n}W_{ij}(x_i - x_j)^2]}{2\sum_{i=1}^{n}\sum_{j=1}^{n}W_{ij}\sum_{i=1}^{n}(x_i - x_j)^2} \tag{2}$$

Geary's C statistic takes values in the range of $[0, 2]$.

The global spatial correlation examines the agglomeration of the whole space, and it is difficult to describe the agglomeration status in the local neighborhood. Therefore, the local Moran's I was used to portray the local spatial agglomeration condition, with the calculation Equation as follows:

$$Local\ Moran's\ I_i = \frac{x_i - \bar{x}}{\frac{1}{n}\sum_{j=1}^{n}(x_i - \bar{x})^2}\sum_{j=1}^{n}W_{ij}(x_j - \bar{x}) \tag{3}$$

**4.1.2. Spatial econometric model.** Combined with theoretical analysis, the spatial econometric model is more appropriate. Current spatial econometrics mainly includes the SDM, SEM, and SAR, all of which have different applicable environments and different assumptions. SDM, a superior model, is a combination of SEM and SAR that comprehensively considers the interdependence between regions and the correlation of disturbance terms. Advantages of the spatial Durbin model: SDM models are able to capture both spatial lag and spatial error terms, it not only has the characteristics of the general spatial measurement model, but also can decompose the direct effects and indirect effects from the total effects.

$$\ln Y_{it} = \ln \alpha + \rho W Y_{it} + \sum \beta_n \ln X_{nit} + \sum \theta_n W \ln X_{nit} + \mu_i + \varepsilon_{it} \tag{4}$$

$$\begin{aligned} \ln Y_{it} &= \ln \alpha + \sum \beta_n \ln X_{nit} + \mu_i + \psi_{it} \\ \psi_{it} &= \lambda W \psi_{it} + \varepsilon_{it} \end{aligned} \tag{5}$$

$$\ln Y_{it} = \ln \alpha + \rho W Y_{it} + \sum \beta_n \ln X_{nit} + \mu_i + \varepsilon_{it} \tag{6}$$

Models (4), (5), and (6) represent SDM, SEM, and SAR, respectively. Y: the dependent variable, X: the independent variable. $\beta_n$: the lasticity coefficient of the independent variable, $\theta_n$: the elasticity coefficient of the spatial lag term of the explanatory variable. $\rho$ the spatial autocorrelation coefficient, $\psi$: the spatial autocorrelation error term, $\lambda$: the spatial autocorrelation

coefficient of the error term, $\alpha$ represents a constant, $\mu_i$: the individual effect, $\varepsilon_{it}$ represents the error term.

### 4.1.3. Panel threshold model

The panel threshold model was applied to verify the nonlinear impact of green technology innovation on carbon emission reduction capacity, Threshold studies mainly include group test, cross-term test and panel threshold model. Group test relies on artificial setting of segmentation points, with strong subjectivity. Cross-term test has uncertainty, and the significance of the estimated results is difficult to solve. The panel threshold model makes up for the shortcomings of the above methods by using the estimated threshold parameter as a segment function, which can automatically identify the mutation points of the data and conduct the significance test. With the model structure being set as follows:

$$\ln Y = \alpha_0 + \alpha_1 \ln X + \beta_1 \ln R*I(Z \leq \gamma) + \beta_2 \ln R*I(Z > \gamma) + \varepsilon \tag{7}$$

$$\ln Y = \alpha_0 + \alpha_1 \ln X + \beta_1 \ln R*I(Z \leq \gamma_1) + \\ \beta_2 \ln R*I(\gamma_1 < Z \leq \gamma_2) + \beta_3 \ln R*I(Z > \gamma_2) + \varepsilon \tag{8}$$

$$\ln Y = \alpha_0 + \alpha_1 \ln X + \beta_1 \ln R*I(Z \leq \gamma_1) + \beta_2 \ln R*I(\gamma_1 < Z \leq \gamma_2) \\ + \beta_3 \ln R*I(\gamma_2 < Z \leq \gamma_3) + \beta_4 \ln R*I(Z > \gamma_4) + \varepsilon \tag{9}$$

Where R represents the threshold dependent variable, and $\gamma$ represent the threshold value to be estimated, respectively. Z represents the threshold variable, $\varepsilon$ represents the random disturbance term.

### 4.2. Variable selection

The explained variable is carbon emission reduction capacity (Cer). According to previous studies [49, 50], a regional carbon emission reduction capacity assessment index system was build based on the basic features of carbon emission reduction capacity and data availability, which contains 30 indicators in 9 dimensions like economic dynamics, energy consumption, carbon sink capacity, carbon emission level, industrial upgrading, carbon transfer capacity, carbon market vitality, carbon governance capacity, and social development capacity, as shown in Table 1. Then the genetic algorithm-projection pursuit model was used to measure the carbon emission reduction capacity.

Independent variable: green technology innovation (*Git)*. Patent data, an effective measure of technology innovation capacity [51], which was characterizated by the number of green invention patents obtained.

Control variables were selected to alleviate the bias caused by omitted variables in this study according to previous studies. It contains: labor productivity (*Labor*) was measured by the GDP created per unit of employed people; population size (*Pop*) was measured by the population per unit area; land area (*Area*) was characterized by the area of the administrative division of each province and city; and fiscal expenditure (*Gov*) was measured by the fiscal expenditure fiscal revenue.

Threshold variables were selected as green technology innovation.

### 4.3. Data sources

This paper takes Chinese provinces as the research object to study the relationship between green science and technology innovation and carbon emission reduction capacity. Select the

**Table 1. Carbon emission reduction capability evaluation indicators.**

| Dimension | Index | Direction |
|---|---|---|
| Economic dynamics | Per capita GDP | + |
| | GDP growth rate | + |
| | Regional GDP/National | + |
| | Per capita net income of rural residents | + |
| | Fixed assets investment | + |
| | Total import and export of goods by foreign-invested enterprises | + |
| | Per capita disposable income of urban residents | + |
| Energy Consumption | Coal as a percentage of total energy consumption | — |
| | Energy consumption per 10000 yuan of GDP | — |
| | Per capita energy consumption | — |
| Carbon sink Capacity | Forest cover | + |
| | Green coverage rate in built-up areas | + |
| | Carbon absorption of crops | + |
| | Forest carbon absorption | + |
| | Per capita park green area | + |
| Carbon emission Level | Carbon emissions/GDP | — |
| | Per capita carbon emissions | — |
| Industrial upgrading | Tertiary sector of the economy/Secondary sector of the economy | + |
| | Industrial rationalization | + |
| Carbon transfer Capacity | Import and Export/GDP | + |
| | FDI/GDP | + |
| Carbon market Vitality | Marketization index | + |
| Carbon governance Capacity | Investment/GDP completed for pollution control | + |
| | Technology Market Turnover/GDP | + |
| Social development Capacity | Ownership of civilian vehicles | + |
| | Consumption level of urban residents | + |
| | Urban gas penetration rate | + |
| | College Student Teacher Ratio | — |
| | Vehicle ownership in highway operation | + |
| | Consumption level of rural residents | + |

macro data of each province. The government agencies collect a large amount of data in the process of performing their duties, and this paper obtains through the government open data platform, public database and other ways. Details are as follows: Data of 30 provinces in China from 2006 to 2021 were extracted in this study. These data were obtained from China Energy Statistical Yearbook, and China's Environmental Yearbook et al., with the missing values being filled with the mean value. The descriptive statistics are shown in Table 2.

## 5. Empirical results and analysis

### 5.1. Carbon emission reduction capacity measurement results and spatial-temporal evolution

After the measurement of the carbon emission reduction capacity, which shown in Table 3. The mean value of China's carbon emission reduction capacity score rises from 1.368 to 2.271 in 2021, with Beijing, Shanghai, and Guangdong ranking the top three, with a combined score of 2.912, 2.701, and 2.561, respectively, of which Beijing and Shanghai are the super-provinces. Ningxia, Guizhou, and Gansu are in the bottom three, with scores of 1.174, 1.265, and 1.294 respectively.

**Table 2. Descriptive statistical results.**

| Variable | Obs | Mean | Std. Dev. | Min | Max |
|---|---|---|---|---|---|
| *Cer* | 480 | 1.843 | .516 | .697 | 3.481 |
| *Git* | 480 | 754.296 | 1270.601 | 1 | 9077 |
| *Labor* | 480 | 8.931 | 6.231 | 1.197 | 53.907 |
| *Pop* | 480 | 4430.572 | 2726.218 | 285 | 12684 |
| *Area* | 480 | 287000 | 359000 | 6340.5 | 1660000 |
| *Gov* | 480 | 2.477 | 1.691 | 1.052 | 15.625 |

In order to examine the spatial variability of the carbon emission reduction capacity and display the measured carbon emission reduction capacity in the graph, four representative years, 2006, 2011, 2016, and 2021, were selected to plot the spatial distribution map according to the quintile method. In addition, the spatial distribution characteristics are observed with references to Table 3. During the observation period, the carbon emission reduction capacity

**Table 3. Comprehensive score of carbon emission reduction capability.**

| Province(City) | 2006 | 2011 | 2016 | 2021 | mean | Rank |
|---|---|---|---|---|---|---|
| Beijing | 2.305 | 2.652 | 3.340 | 3.393 | 2.912 | 1 |
| Tianjin | 1.777 | 2.089 | 2.491 | 2.400 | 2.192 | 6 |
| Hebei | 1.235 | 1.588 | 1.883 | 2.115 | 1.713 | 20 |
| Shanxi | 0.824 | 1.256 | 1.492 | 1.634 | 1.319 | 26 |
| Inner Mongolia | 1.100 | 1.376 | 1.719 | 1.718 | 1.485 | 24 |
| Liaoning | 1.567 | 1.946 | 1.972 | 2.125 | 1.943 | 9 |
| Jilin | 1.242 | 1.550 | 1.816 | 2.003 | 1.673 | 21 |
| Heilongjiang | 1.277 | 1.622 | 1.919 | 2.057 | 1.737 | 18 |
| Shanghai | 2.129 | 2.587 | 3.030 | 3.187 | 2.701 | 2 |
| Jiangsu | 1.737 | 2.202 | 2.670 | 2.997 | 2.375 | 5 |
| Zhejiang | 1.822 | 2.271 | 2.727 | 3.015 | 2.438 | 4 |
| Anhui | 1.219 | 1.590 | 1.933 | 2.286 | 1.754 | 16 |
| Fujian | 1.783 | 2.006 | 2.399 | 2.579 | 2.178 | 7 |
| Jiangxi | 1.444 | 1.800 | 2.080 | 2.360 | 1.908 | 10 |
| Shandong | 1.570 | 1.899 | 2.306 | 2.540 | 2.052 | 8 |
| Henan | 1.153 | 1.528 | 1.952 | 2.303 | 1.734 | 19 |
| Hubei | 1.358 | 1.628 | 2.049 | 2.377 | 1.849 | 13 |
| Hunan | 1.374 | 1.659 | 2.018 | 2.378 | 1.850 | 12 |
| Guangdong | 2.157 | 2.343 | 2.790 | 3.085 | 2.561 | 3 |
| Guangxi | 1.298 | 1.629 | 1.919 | 2.143 | 1.751 | 17 |
| Hainan | 1.471 | 1.729 | 2.003 | 2.243 | 1.833 | 15 |
| Chongqing | 1.230 | 1.744 | 2.058 | 2.315 | 1.843 | 14 |
| Sichuan | 1.340 | 1.718 | 2.023 | 2.436 | 1.864 | 11 |
| Guizhou | 0.697 | 1.122 | 1.530 | 1.878 | 1.315 | 27 |
| Yunnan | 1.074 | 1.489 | 1.731 | 2.016 | 1.589 | 23 |
| Shaanxi | 1.214 | 1.508 | 1.840 | 2.089 | 1.668 | 22 |
| Gansu | 0.873 | 1.081 | 1.440 | 1.691 | 1.265 | 29 |
| Qinghai | 1.015 | 1.206 | 1.406 | 1.409 | 1.294 | 28 |
| Ningxia | 0.710 | 1.016 | 1.408 | 1.594 | 1.174 | 30 |
| Xinjiang | 1.055 | 1.246 | 1.396 | 1.780 | 1.333 | 25 |
| Mean | 1.368 | 1.703 | 2.045 | 2.271 | 1.843 | / |

**Table 4. The results of stability test.**

| Variable | LLC | | IPS | | Conclusion |
|---|---|---|---|---|---|
| | Statistical | P | Statistical | P | |
| LnCer | -6.3115*** | 0.0000 | -4.0871*** | 0.0000 | Stability |
| lnGit | -1.8599** | 0.0314 | -5.8134*** | 0.0000 | Stability |
| lnLabor | -2.6920*** | 0.0036 | -1.7146** | 0.0432 | Stability |
| lnPop | -2.7855*** | 0.0027 | -1.7836** | 0.0372 | Stability |
| lnArea | -2.5468*** | 0.0054 | -1.3522* | 0.0082 | Stability |
| lnGov | -5.0582*** | 0.0000 | -3.2506*** | 0.0006 | Stability |

Note

*, **, and ***respectively represent significant levels at 10%, 5%, and 1%, the same below.

shows a gradual upward trend over time in general. However, there are still large differences. Beijing and Shanghai as the super-provinces that are at the forefront every year. In conclusion, the spatial layout shows the situation of "leading in eastern China, rising of central China, and catching up with western China".

## 5.2. Stationarity test

LLC and IPS methods were selected for the Stationarity test, which shown in Table 4. Variables were not consistent with those of the original hypothesis and were considered stable data.

## 5.3. Multicollinearity

The high degree of correlation between two or more variables will lead to instability in the analysis of the results. Therefore, VIF was used to measure the degree of multicollinearity to verify whether there is multicollinearity between explanatory variables, with results shown in Table 5. It can be found that the maximum of VIF is 4.080. The mean of VIF is 3.180, indicating that there is no need to exclude the study variables as there is no serious collinearity.

## 5.4. Spatial spillover effect

**5.4.1. Spatial correlation analysis.** The presupposes of adopting spatial model is the variables have spatial correlation, so first need to inspect the spatial correlation. Geary's C and Moran's I were applied in this study to test, which shown in Tables 6 and 7.

From Tables 6 and 7, it can be found that except for very few years, the global Geary's C indexes of carbon them are both positive, while Moran's I indexes are both positive, which indicates that a positive spatial correlation between the them.

**Table 5. The results of multicollinearity test.**

| Variable | VIF | 1/VIF |
|---|---|---|
| lnGov | 4.080 | 0.245 |
| lnPop | 3.200 | 0.312 |
| lnGit | 2.960 | 0.338 |
| lnArea | 2.880 | 0.347 |
| lnLabor | 2.790 | 0.358 |
| Mean VIF | 3.180 | |

**Table 6. Global Geary's C and Moran's I of carbon reduction capability.**

| Year | Geary's C | Z | P | Moran's I | Z | P |
|------|-----------|--------|-------|-----------|-------|-------|
| 2006 | 0.661 | -2.939 | 0.002 | 0.303 | 3.183 | 0.001 |
| 2007 | 0.642 | -3.124 | 0.001 | 0.304 | 3.187 | 0.001 |
| 2008 | 0.605 | -3.472 | 0.000 | 0.333 | 3.454 | 0.000 |
| 2009 | 0.594 | -3.564 | 0.000 | 0.349 | 3.606 | 0.000 |
| 2010 | 0.591 | -3.596 | 0.000 | 0.338 | 3.498 | 0.000 |
| 2011 | 0.618 | -3.349 | 0.000 | 0.326 | 3.388 | 0.000 |
| 2012 | 0.622 | -3.318 | 0.000 | 0.034 | 3.407 | 0.000 |
| 2013 | 0.600 | -3.517 | 0.000 | 0.345 | 3.568 | 0.000 |
| 2014 | 0.605 | -3.468 | 0.000 | 0.346 | 3.578 | 0.000 |
| 2015 | 0.614 | -3.384 | 0.000 | 0.335 | 3.476 | 0.000 |
| 2016 | 0.578 | -3.685 | 0.000 | 0.367 | 3.783 | 0.000 |
| 2017 | 0.606 | -3.435 | 0.000 | 0.343 | 3.555 | 0.000 |
| 2018 | 0.678 | -2.800 | 0.003 | 0.296 | 3.115 | 0.001 |
| 2019 | 0.721 | -2.413 | 0.008 | 0.259 | 2.772 | 0.003 |
| 2020 | 0.766 | -2.041 | 0.021 | 0.217 | 2.376 | 0.009 |
| 2021 | 0.719 | -2.457 | 0.007 | 0.237 | 2.554 | 0.005 |

Next, continued to carry out the local Moran test on the representative years 2006, 2011, 2016, and 2021, with the results shown in Figs 1–8, which showed that both carbon emission reduction capacity and green innovation have significant H-H and L-L characteristics. In the displayed years, the provinces and cities whose carbon emission reduction capacity in the first or third quadrants account for 66.7%, 76.7%, 76.7%, and 60% of the provinces, whose green technology innovation in the first or third quadrants account for 70%, 70%, 60%, and 60% of the provinces and cities, showing the strong spatial agglomeration effect of carbon emissions reduction capacity and green technology innovation. One side, the H-H agglomeration areas of carbon emission reduction capacity are mainly distributed in Beijing, Tianjin, Shanghai, while the L-L areas are mainly distributed in Shaanxi and Xinjiang. On the other hand, the

**Table 7. Global Geary's C and Moran's I of green technology innovation.**

| Year | Geary's C | Z | P | Moran's I | Z | P |
|------|-----------|--------|-------|-----------|-------|-------|
| 2006 | 0.760 | -2.007 | 0.022 | 0.184 | 2.906 | 0.018 |
| 2007 | 0.759 | -2.110 | 0.017 | 0.225 | 2.439 | 0.007 |
| 2008 | 0.755 | -2.149 | 0.016 | 0.218 | 2.378 | 0.009 |
| 2009 | 0.834 | -1.444 | 0.074 | 0.159 | 1.831 | 0.034 |
| 2010 | 0.859 | -1.161 | 0.123 | 0.117 | 1.468 | 0.071 |
| 2011 | 0.794 | -1.784 | 0.037 | 0.176 | 1.987 | 0.023 |
| 2012 | 0.783 | -1.887 | 0.030 | 0.184 | 2.066 | 0.019 |
| 2013 | 0.809 | -1.621 | 0.052 | 0.140 | 1.667 | 0.048 |
| 2014 | 0.796 | -1.685 | 0.046 | 0.149 | 1.775 | 0.038 |
| 2015 | 0.813 | -1.625 | 0.052 | 0.146 | 1.700 | 0.045 |
| 2016 | 0.804 | -1.713 | 0.043 | 0.152 | 1.754 | 0.040 |
| 2017 | 0.835 | -1.445 | 0.074 | 0.135 | 1.591 | 0.056 |
| 2018 | 0.838 | -1.428 | 0.077 | 0.130 | 1.545 | 0.061 |
| 2019 | 0.857 | -1.260 | 0.104 | 0.115 | 1.397 | 0.081 |
| 2020 | 0.843 | -1.396 | 0.081 | 0.126 | 1.503 | 0.066 |
| 2021 | 0.791 | -1.848 | 0.032 | 0.169 | 1.910 | 0.028 |

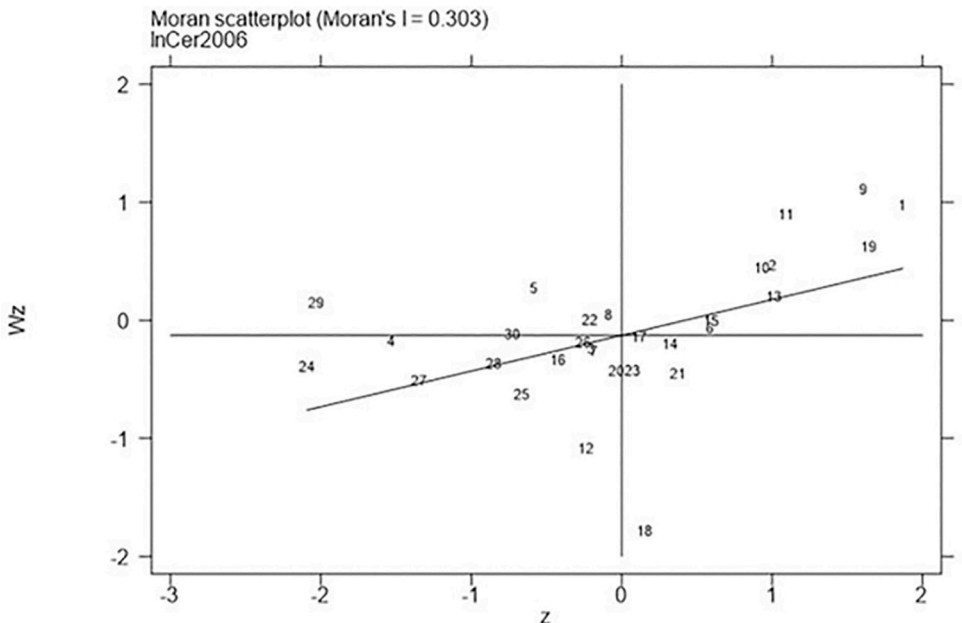

**Fig 1. Moran scatter plot of carbon emission reduction capacity (Year 2006).**

H-H agglomeration areas of green technology innovation are mainly distributed in Beijing, Jiangsu, Zhejiang, Chongqing, Fujian, while the L-L areas are mainly distributed in Guizhou, Yunnan, Gansu, Xinjiang. Therefore, the SDM was used for the next empirical analysis.

**5.4.2. Spatial econometric model test.** SDM, SEM, and SAR are all common spatial econometric models, but what kind of model is used needs to be tested, that is, to determine

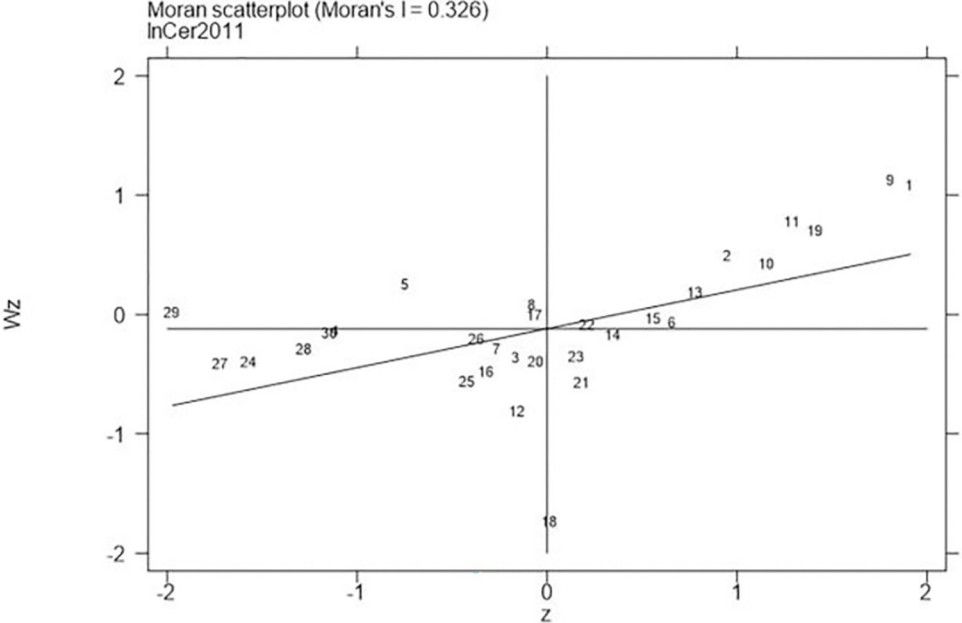

**Fig 2. Moran scatter plot of carbon emission reduction capacity (Year 2011).**

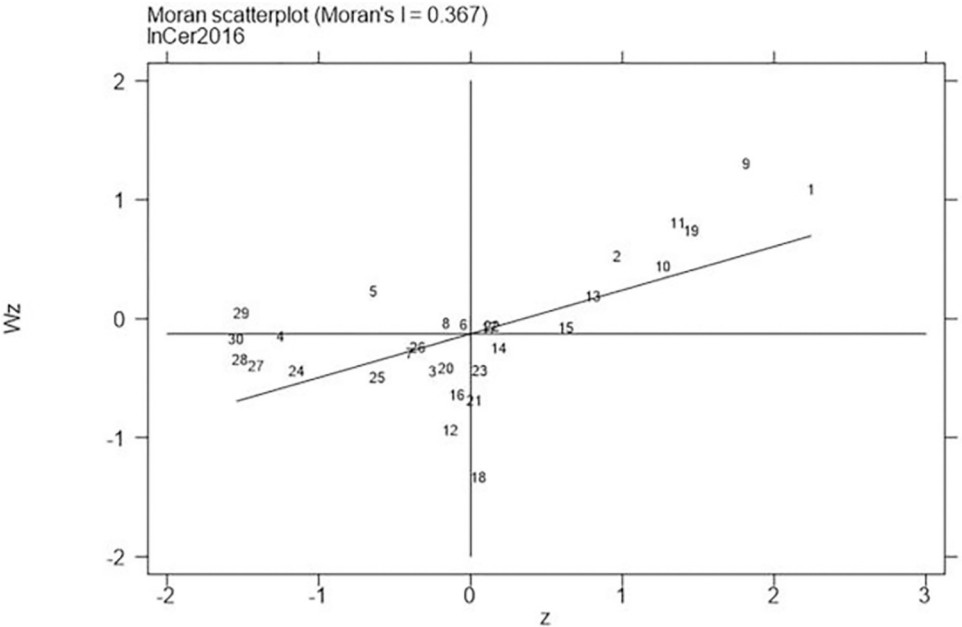

**Fig 3. Moran scatter plot of carbon emission reduction capacity (Year 2016).**

whether the SDM is degraded to the SEM and SAR, with the test results shown in Table 8. The results showed it was most appropriate to apply the SDM as the LR and the Wald test.

**5.4.3. Analysis results of SDM.** In order to accurately estimate the real impact of green technology innovation on carbon emission reduction capacity, the effect was split into direct and spillover effect based on the decomposition perspective of the SDM. The results are listed in Table 9. From the point of view of the direct effect, the estimated coefficient of green

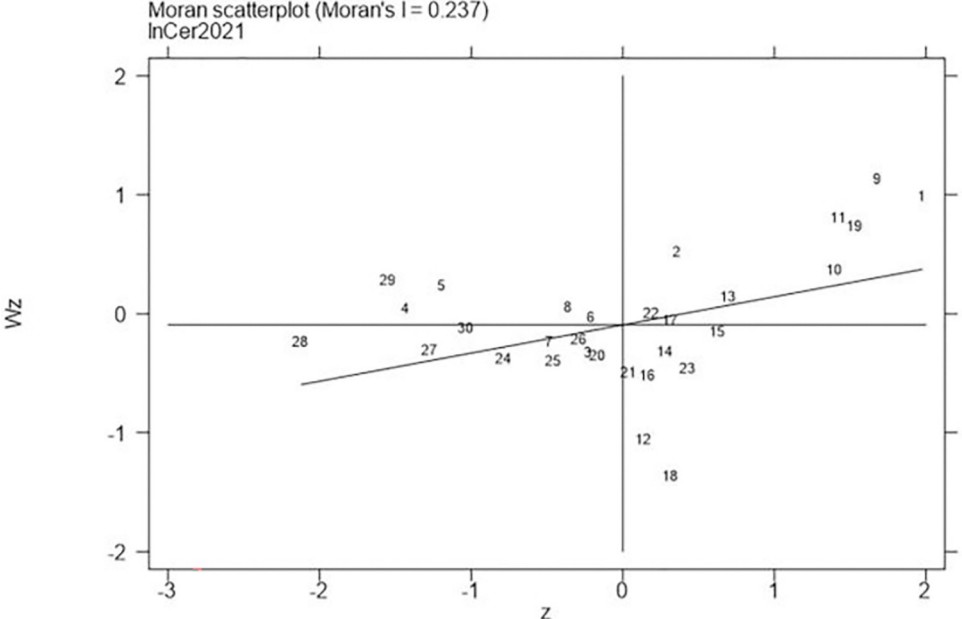

**Fig 4. Moran scatter plot of carbon emission reduction capacity (Year 2021).**

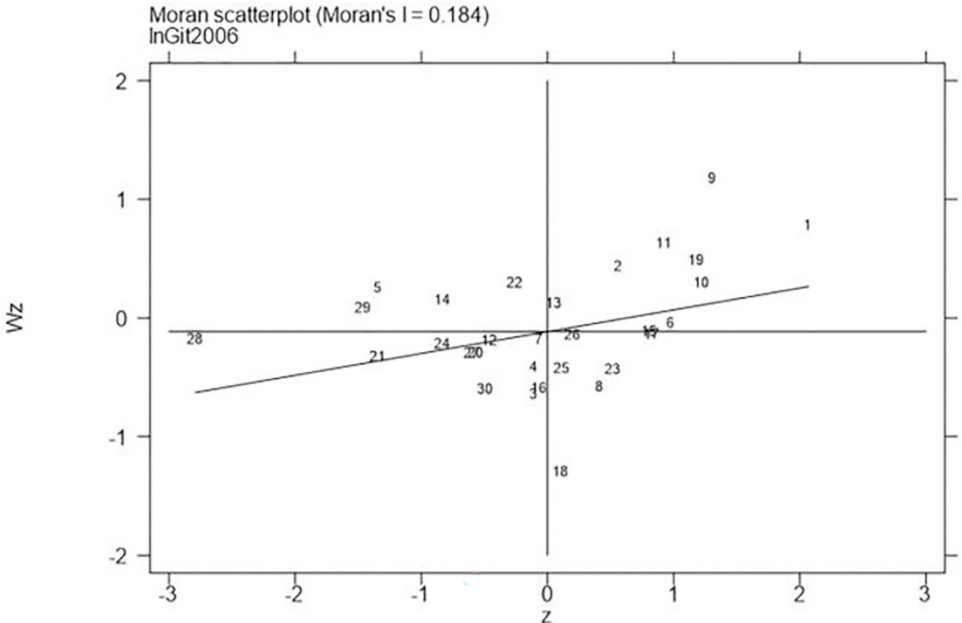

**Fig 5. Moran scatter plot of green technology innovation (Year 2006).**

technology innovation was 0.021, indicating that green innovation significantly promoted the enhancement of carbon emission reduction capacity and verified Hypothesis H1. Which agree with the view of Tobelmann [52].

From the view of spillover effect, the estimated coefficient was 0.076, making known green technology innovation had a significant positive effect on carbon emission reduction capacity, which verified Hypothesis H2.while green innovation has technology spillover effect [53], which can inhibit neighboring carbon emissions [54].

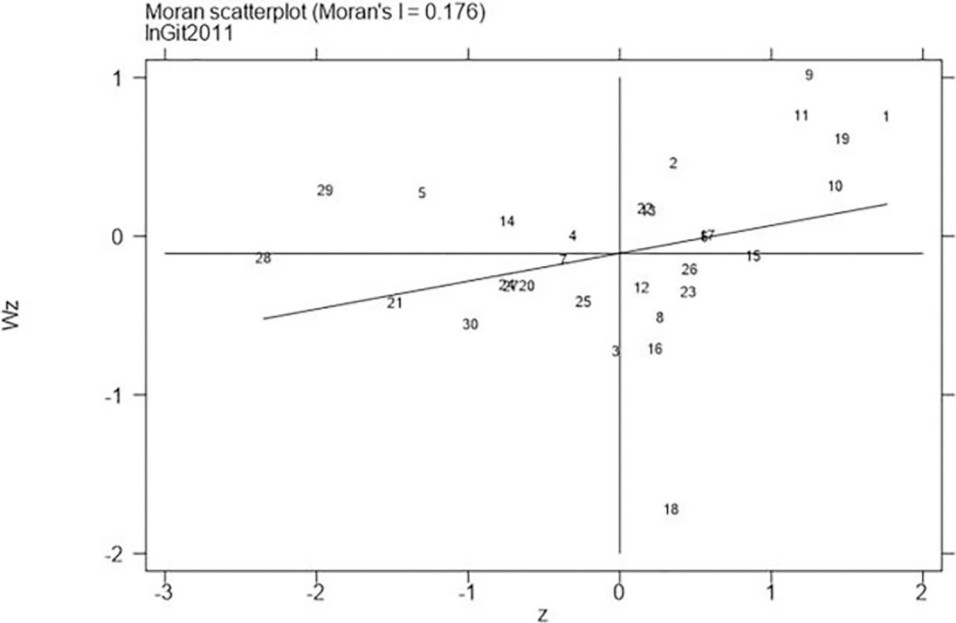

**Fig 6. Moran scatter plot of green technology innovation (Year 2011).**

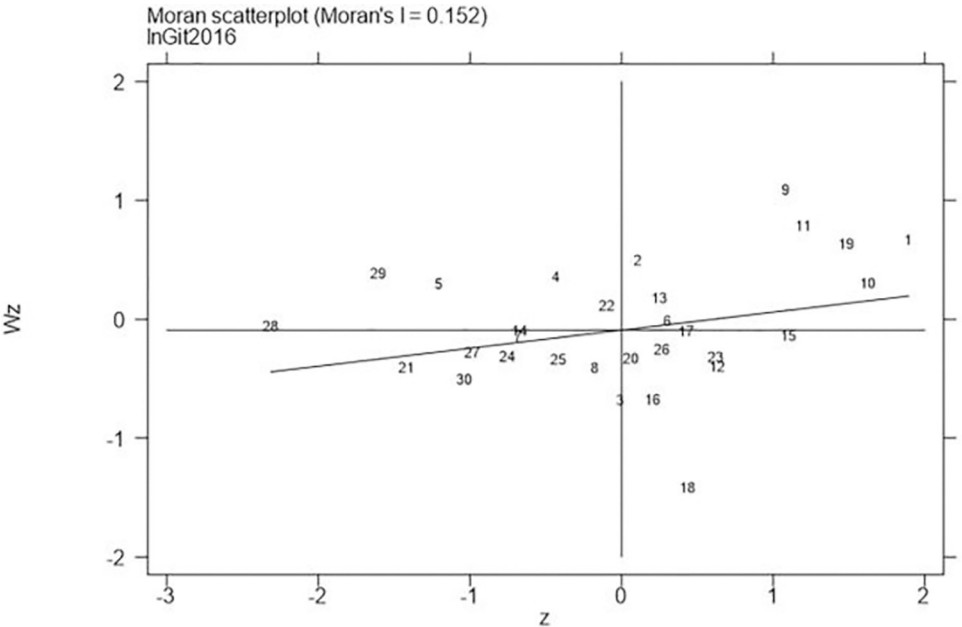

**Fig 7. Moran scatter plot of green technology innovation (Year 2016).**

**5.4.4. Robustness test.** The robustness test was performed to exclude the possibility that the study results were not completely correct. Replacement variables are one of the most common robustness test methods. Different ways of measuring variables may lead to deviation in the results. Spatial measurement model is sensitive to the setting of the relationship of spatial weight matrix, and different matrices may also lead to deviation in the results. Therefore, to guarantee the dependability of the above findings, a series of methods were used for robustness

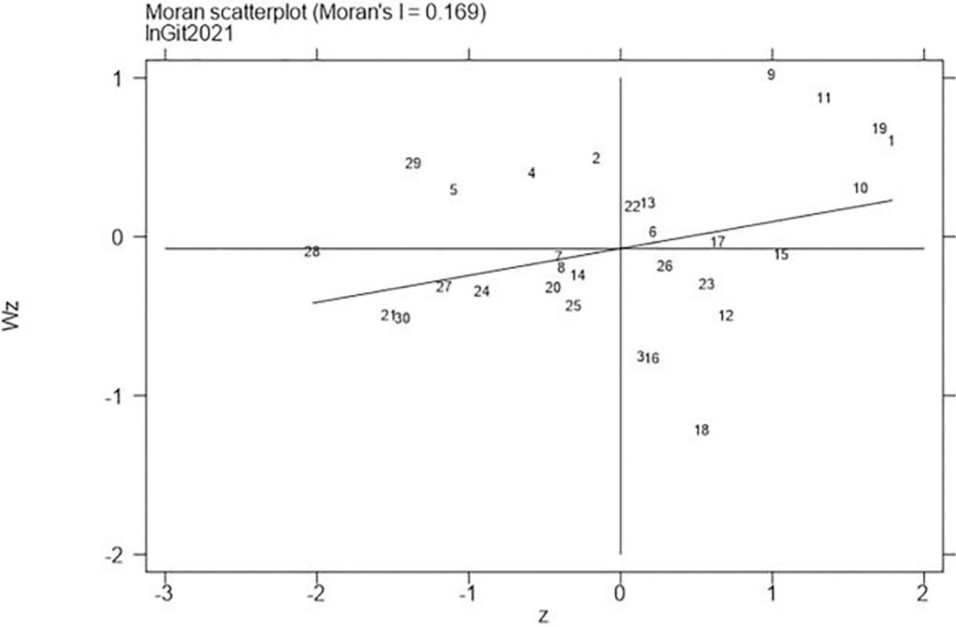

**Fig 8. Moran scatter plot of green technology innovation (Year 2021).**

**Table 8. Inspection results of space panel model.**

| Inspection type | Verification model | St | P | Sig |
|---|---|---|---|---|
| Wald | SAR | 44.76 | 0.0000 | *** |
|  | SEM | 100.86 | 0.0000 | *** |
| LR | SAR | 42.96 | 0.0000 | *** |
|  | SEM | 172.46 | 0.0000 | *** |

testing. (1) Replacement of core explanatory variables method. The core explanatory variables were replaced with the number of green inventions obtained per capita as substitution variable; (2) The spatial weight matrix was replaced with a combined economic and geographic matrix. The robustness results are shown in Table 10, indicating that the estimated coefficient of green technology innovation on carbon emission reduction capacity have slight differences whether replacing the core explanatory variables or the matrix, with the sign of coefficients, the significance level being consistent.

## 5.5. Threshold characteristics

**5.5.1. Threshold existence test.**   The panel threshold model was used to test whether there is a nonlinearity between them, the samples were repeatedly extracted through the Bootstrap method to conduct the threshold existence test. Firstly, to test whether single threshold or not, and in turn, to test whether there is a double threshold or triple threshold, with the results shown in Table 11. It was found double thresholds between them. Therefore, there are two threshold values.

**5.5.2. Threshold truthfulness test.**   Whether the threshold is real or not can be judged by the confidence interval and the graph of the LR likelihood ratio function, the results of the threshold truthfulness test shown in Table 12 and Fig 9 which can be found from the table that the threshold estimations were 3.6636 and 4.7185, and the corresponding confidence intervals were [3.6106, 3.6889] and [4.6578, 4.7274], respectively which shows that the threshold estimations are valid and pass the truthfulness test, there is a double threshold model.

**Table 9. Direct and spillover effects.**

| Effect category | Variable | Coefficient | Standard Error | T | P | 95% confidence interval | |
|---|---|---|---|---|---|---|---|
| LR_Direct | lnGit | 0.021*** | 0.007 | 2.820 | 0.005 | 0.006 | 0.036 |
|  | lnLabor | 0.031*** | 0.012 | 2.660 | 0.008 | 0.008 | 0.054 |
|  | lnPop | 0.018* | 0.010 | 1.760 | 0.079 | -0.002 | 0.038 |
|  | lnArea | -0.014*** | 0.004 | -3.200 | 0.001 | -0.023 | -0.005 |
|  | lnGov | 0.066*** | 0.018 | 3.760 | 0.000 | 0.032 | 0.101 |
| LR_Indirect | lnGit | 0.076*** | 0.011 | 6.740 | 0.000 | 0.054 | 0.098 |
|  | lnLabor | 0.090*** | 0.022 | 4.080 | 0.000 | 0.047 | 0.133 |
|  | lnPop | -0.006 | 0.040 | -0.160 | 0.874 | -0.085 | 0.072 |
|  | lnArea | -0.002 | 0.015 | -0.140 | 0.891 | -0.032 | 0.028 |
|  | lnGov | 0.163*** | 0.041 | 4.000 | 0.000 | 0.083 | 0.242 |
| LR_Total | lnGit | 0.097*** | 0.011 | 9.010 | 0.000 | 0.076 | 0.118 |
|  | lnLabor | 0.121*** | 0.023 | 5.290 | 0.000 | 0.076 | 0.166 |
|  | lnPop | 0.012 | 0.047 | 0.250 | 0.802 | -0.080 | 0.104 |
|  | lnArea | -0.016 | 0.016 | -0.980 | 0.327 | -0.048 | 0.016 |
|  | lnGov | 0.229*** | 0.045 | 5.100 | 0.000 | 0.141 | 0.317 |

**Table 10. Robustness test results.**

| Effect category | Replace core explanatory variables | Replace matrix |
|---|---|---|
| LR_Direct | 0.021*** | 0.068*** |
| | (0.01) | (0.01) |
| LR_Indirect | 0.076*** | 0.033*** |
| | (0.01) | (0.01) |
| LR_Total | 0.097*** | 0.101*** |
| | (0.01) | (0.01) |

**Table 11. Threshold existence test.**

| Threshold type | Fstat | Prob | BS Count | Crit1 | Crit5 | Crit10 |
|---|---|---|---|---|---|---|
| Single threshold | 29.06* | 0.0867 | 300 | 28.0762 | 32.7605 | 43.9523 |
| Double threshold | 25.59** | 0.0333 | 300 | 19.2427 | 24.3672 | 29.4468 |
| Triple threshold | 4.78 | 0.6433 | 300 | 15.1806 | 18.4938 | 24.7402 |

**Table 12. Threshold estimates.**

| Model | Threshold | Lower | Upper |
|---|---|---|---|
| Single threshold | 4.7185 | 4.6578 | 4.7274 |
| Double threshold | 3.6636 | 3.6106 | 3.6889 |

**Table 13. Threshold estimation results.**

| Variable or Parameter | Coef. | Std. Err. | t | P>|t| | [95% Conf. Interval] | |
|---|---|---|---|---|---|---|
| $lnLabor$ | 0.130*** | 0.011 | 12.220 | 0.000 | 0.109 | 0.151 |
| $lnPop$ | 0.059*** | 0.009 | 6.290 | 0.000 | 0.040 | 0.077 |
| $lnArea$ | -0.030*** | 0.005 | -6.100 | 0.000 | -0.040 | -0.020 |
| $lnGov$ | 0.170*** | 0.018 | 9.610 | 0.000 | 0.135 | 0.204 |
| $lnGit<3.6636$ | 0.046*** | 0.008 | 5.630 | 0.000 | 0.030 | 0.062 |
| $3.6636 \leq lnGit < 4.7185$ | 0.063*** | 0.007 | 9.560 | 0.000 | 0.050 | 0.076 |
| $4.7185 \leq lnGit$ | 0.074*** | 0.006 | 12.650 | 0.000 | 0.062 | 0.085 |
| _cons | -0.329*** | 0.067 | -4.880 | 0.000 | -0.462 | -0.197 |
| N | 480 | | | | | |
| Number of id | 30 | | | | | |
| within $R^2$ | 0.8845 | | | | | |

**5.5.3. Results analysis of threshold regression model.** The results are shown in Table 13, indicating that there is a significant nonlinear characteristic of them under different threshold. Specifically, the impact of green technology innovation on carbon emission reduction capacity is positive when the threshold is less than 3.6636, with an estimated coefficient of 0.046, which will rise to 0.063 when threshold is between 3.6636 and 4.7185. Rise to 0.074 when the green technology innovation is greater than the second threshold value. This indicates that relationship between them showed a gradual increase in non-linear characteristics, which verified Hypothesis H3. Which is basically consistent with the view of Liu et al. [55]. When the green

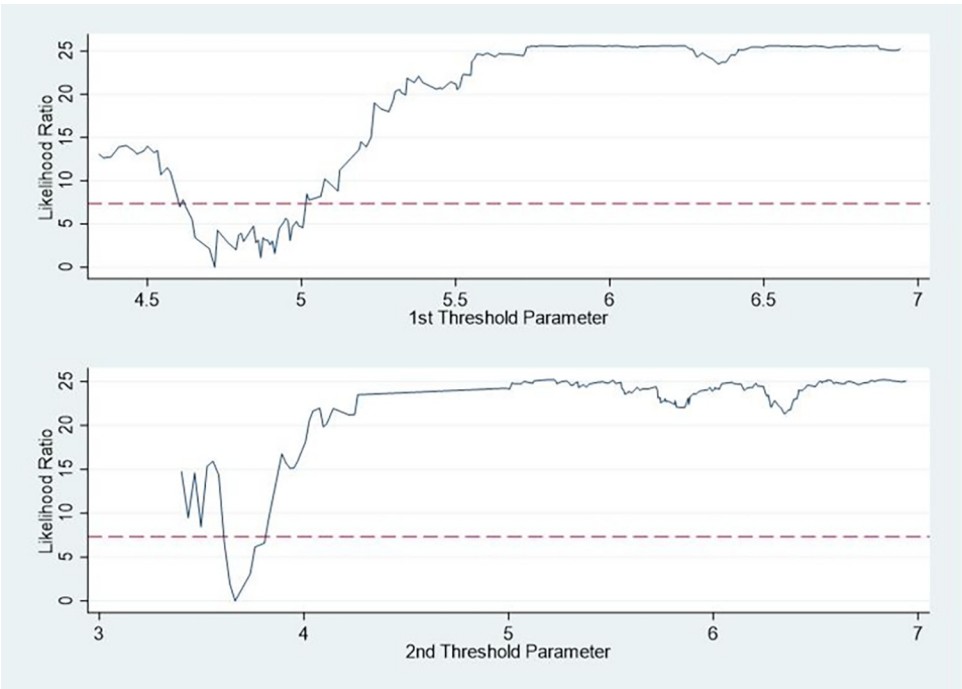

**Fig 9. LR likelihood ratio function diagram.**

innovation ability is weak, the carbon emission reduction effect of green innovation is weak, while when the green innovation is strong, the carbon emission reduction effect is more significant.

**5.5.4 Robustness test.** In order to verify the robustness of the threshold effect, the added control variable method was used to test and increase the government intervention (financial expenditure / GDP) variable. The results are shown in Table 14. The results found that the threshold value and the estimated coefficient were consistent with the original conclusion, with only small differences in the coefficient, indicating that the study results were more reliable.

**Table 14. Results of the robustness test of the threshold effect.**

| Variable or Parameter | Coef. | Std. Err. | t | P>|t| | [95% Conf. Interval] | |
|---|---|---|---|---|---|---|
| lnLabor | 0.133*** | 0.011 | 12.370 | 0.000 | 0.112 | 0.154 |
| lnPop | 0.065*** | 0.010 | 6.510 | 0.000 | 0.045 | 0.085 |
| lnArea | -0.029*** | 0.005 | -5.800 | 0.000 | -0.039 | -0.019 |
| lnGov | -0.151*** | 0.021 | -7.250 | 0.000 | -0.191 | -0.110 |
| lnFis | 0.035* | 0.020 | 1.750 | 0.082 | -0.004 | 0.074 |
| lnGit < 3.6636 | 0.043*** | 0.008 | 5.200 | 0.000 | 0.027 | 0.059 |
| 3.6636 ≤ lnGit < 4.7185 | 0.059*** | 0.007 | 8.470 | 0.000 | 0.045 | 0.073 |
| 4.7185 ≤ lnGit | 0.06*** | 0.006 | 10.890 | 0.000 | 0.057 | 0.082 |
| _cons | -0.308*** | 0.068 | -4.510 | 0.000 | -0.442 | -0.174 |
| N | 480 | | | | | |
| Number of id | 30 | | | | | |
| within R2 | 0.8853 | | | | | |

## 6. Discussion

In terms of the measurement results of carbon emission reduction capacity, which of each province in China shows an increasing trend, which is in line with the reality that each region enters a critical period in which carbon reduction is the key strategic direction because of the acceleration of the pace of ecological civilization construction. In such a critical period, Beijing and Shanghai are super-provinces in reducing carbon emissions, with the overall trend of "leading in eastern China, rising of central China, and catching up with western China", that is basically consistent with the viewpoint of Yao [56].

In terms of the direct effect, the coefficient of green innovation was 0.021, which verifies Hypothesis H1. The possible reasons for this situation lie in: On the one hand, the carbon emission problem in some areas is getting worse because of the weak innovation consciousness and innovation basis. Progress in Green Innovation provides these areas with green and low-carbon development ideas, thus promoting various types of carbon emission implementers to make carbon emission reduction driven by innovation through effective technical support. In short, green technology innovation reduces the loss of fossil fuels, thus suppressing carbon emissions [57]. The other side, innovation is the main enterprise, associated with the government, industry, scientific research institutions, and other relevant departments. As we all know, enterprises are oriented to consumer demand. With the concept of sustainable consumption deeply rooted in people's hearts, consumers began to pursue green products, which are closely related to green innovation design, green product and service supply, and green process innovation, all of which undoubtedly provide a basis for creating energy-saving and environmentally friendly products. Based on this, traditional industries have been transformed accordingly and emerging industries have become greener, resulting in the large increase of green and low-carbon industries, which in turn greatly reduces $CO_2$ emissions and increase carbon emission reduction capacity.

(3) Green technology innovation has a significant positive spatial spillover effect. The possible reasons for this situation lie in: One side, green technology innovation have the general characteristics of innovation. The spillover of green innovation activities accelerates the dissemination of elements and knowledge results to enterprises. Therefore, enterprises' access to green innovation knowledge will greatly stimulate the interest of regional enterprises in enhancing green innovation, which will show a high level of enthusiasm for carbon emission reduction. On the other side, carbon emission reduction, an important embodiment of people's livelihood concept, has been included in the achievement assessment. The region has shown top-by-top competition due to the green innovation in the neighboring regions. In addition, the government will also pursue the improvement of green technology innovation due to pressure of achievement assessment, which is manifested in the corresponding system of carbon emission reduction, thus appearing to implement the imitation behavior.

(4) There is a double threshold effect in the impact of green technology innovation on carbon emission reduction capacity, which is characterized by a gradual increase in non-linear characteristics. The possible reasons for this situation should be that green technology innovation can promote the green transformation of high-pollution industries and enhancing efficiency of resource utilization. As green technology innovation crosses threshold value, the marginal cost of pollution control and further declines. In addition, local governments and enterprises are more inclined to introduce green technology innovations or even innovate on their own due to the high value-added characteristics of innovations, thus forming a virtuous circle and promoting the further enhancement of carbon emission reduction capacity.

## 7. Research conclusion and insights

### 7.1 Research conclusion

Develop low-carbon economy under the dual-carbon goal is more characteristic of the times,. Based on the measurement of the carbon reduction capacity of 30 provinces in China, SDM and threshold model were constructed in this study to empirically analyze the spatial spillover effect and the impact of different green technology innovations on carbon reduction capacity. The study results show: The carbon emission reduction capacity of China gradually rises with time, showing the distribution characteristics of eastern best, central average, western worse. However, there are large differences in carbon emission reduction capacity between regions, showing an evolutionary trend from narrowing to rising. Green innovation, carbon emission reduction capacity show significant positive spatial correlation, green technology innovation not only promotes carbon emission reduction capacity in the area, simultaneously, it influences carbon emission reduction capacity in the adjacent areas. In addition, green technology innovation has a double threshold effect. On the one side, there are significant differences in the impact of different green technology innovations. The other side, the enhancement of green technology innovation on the carbon emission reduction capacity of each region presents incremental nonlinear characteristics.

### 7.2 Suggestions

Firstly, a new pattern of regional carbon emission reduction "big synergy" should be built. On side, the government should give full play to its macro-coordination ability to establish an executive committee for coordinated management of carbon emissions and pollution. This can break down administrative barriers and divide the carbon reduction responsibilities of different provinces according to the characteristics in each region, and unify the planning of carbon emission regulation mechanisms. On the other side, each government abide by the environmental bottom line and establish a good achievement view to guide the changeover from traditional high-carbon enterprises to low-carbon enterprises. In addition, the neighboring provinces should also change the corresponding supporting industries to improvement the coordination of carbon reduction.

Secondly, strengthened the in-depth integration of green innovation and carbon emission. Carbon emission reduction can only work if green technology innovation is better applied and green technology innovation is matched with carbon emissions. On the one hand, increase the subsidies for scientific and technological research and development to encourage enterprises, research institutes, universities, and other institutions to engage in innovation activities and support green and low-carbon transformation of industries. On the other hand, gradually guiding of consumers to choose green low-carbon products can promote enterprises to optimize resources, thus achieving green low-carbon development.

Thirdly, green and low-carbon differentiated standards should be formulated to promote a comprehensive green transformation. We should gradually narrow the gap in the level of green technology innovation because of the significant differences in green and science technology innovation and carbon emissions. For regions with low green technology innovation, they should capitalize on the spillover characteristics of innovation to play a late-development advantage and catching-up effect. What's more, the central government should tilt all innovation resources to these regions to narrow the gap with provinces with high levels of innovation. However, for regions with high green technology innovation, they should rely on their location advantages to attract more talents and safeguard their leading position.

### 7.3 Limitations

This study is mainly based on the data of China, which has its own characteristics in carbon emissions as the world factory. Therefore, these suggestions are only applicable to the Chinese region and contribute a little to the world although the conclusions obtained from the study are somewhat generalizable. In addition, the pathway through which green technology innovation can contribute to carbon emission reduction has not yet been explored. Therefore, in the next study, a spatial mediation effect model should be constructed based on global data to explore the pathways through which green technology innovation can contribute to carbon emission reduction.

## Author Contributions

**Conceptualization:** Wei Wei.

**Data curation:** Yu Ma.

**Formal analysis:** Yu Ma.

**Methodology:** Wei Wei.

**Resources:** Yu Ma.

**Supervision:** Wei Wei.

**Writing – original draft:** Yu Ma.

**Writing – review & editing:** Wei Wei.

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
