## [Decision Letter · Decision Letter 0]

6 May 2024

PONE-D-23-42805The Impact of Green Technology Innovation on Carbon Emission Reduction Capacity in China: Based on Spatial Econometrics and Threshold Effect AnalysisPLOS ONE

Dear Dr. Ma,

Thank you for submitting your manuscript to PLOS ONE. After careful consideration, we feel that it has merit but does not fully meet PLOS ONE’s publication criteria as it currently stands. Therefore, we invite you to submit a revised version of the manuscript that addresses the points raised during the review process.

We look forward to receiving your revised manuscript.

Kind regards,

Najabat Ali, Ph.D.

Academic Editor

PLOS ONE

Journal Requirements:

3. Thank you for stating the following financial disclosure: "Henan Province Philosophy and Social Science Planning Project (No.:2022BJJ038), Soft Science Research Project of Henan Province (No.:232400410004), The Key Scientific Research Project of Colleges and Universities in Henan Province (No.:23A790032)."

Reviewers' comments:

Reviewer's Responses to Questions

**Comments to the Author**

1. Is the manuscript technically sound, and do the data support the conclusions?

Reviewer #1: Yes

Reviewer #2: Partly

2. Has the statistical analysis been performed appropriately and rigorously? 

Reviewer #1: Yes

Reviewer #2: No

3. Have the authors made all data underlying the findings in their manuscript fully available?

Reviewer #1: Yes

Reviewer #2: No

4. Is the manuscript presented in an intelligible fashion and written in standard English?

Reviewer #1: Yes

Reviewer #2: No

5. Review Comments to the Author

Reviewer #1: It is my pleasure to review the current manuscript having title “The Impact of Green Technology Innovation on Carbon Emission Reduction Capacity in China: Based on Spatial Econometrics and Threshold Effect Analysis” for the esteemed journal. The manuscript has some merits but there are certain issues which need to be addressed to improve the quality of the manuscript. The authors must consider the following suggestions:

1. The abstract is well-structured. However, it should further underscore the scientific value added of your paper in your abstract.

2. The novelty of this paper should be further justified by highlighting main contributions of the paper.

3. More recent and relevant papers should be cited to enrich the literature.

4. In the methodology section, the model of the study needs to be justified by comparing it with other models.

5. In the results section, the obtained results should be compared with existing studies in the field.

6. Grammar check is required to avoid any possible English errors.

Reviewer #2: Dear Autor(s)

Thank you for submitting your manuscript, "The Impact of Green Technology Innovation on Carbon Emission Reduction Capacity in China: Based on Spatial Econometrics and Threshold Effect Analysis," to our journal. This topic is crucial and timely, particularly in the context of global efforts to reduce carbon emissions. Your methodological choice of spatial econometrics and threshold effect analysis is robust and suitable. However, several areas require enhancement to clarify, deepen, and empirically support your manuscript more effectively.

The abstract should more distinctly articulate your unique contributions, setting your work apart from existing studies. In the introduction, explicitly state your research questions and objectives rather than implying them to guide your readers more clearly. Your literature review should extend beyond Chinese studies to include global contexts, enhancing the applicability and depth of your analysis. More critical engagement with the existing literature, highlighting limitations and controversies, will solidify your study's foundation.

Your theoretical framework and hypotheses are promising but need clearer explanations of mechanisms like spatial spillover effects and non-linear characteristics within green technology contexts. Definitions for key terms and variables, such as "green technology innovation," must be precise to maintain scientific rigor.

Your methodology should include detailed descriptions of your data collection processes, sources, and integrity measures. Clarify how key variables are measured and operationalized and provide detailed justifications for your chosen statistical methods and models, including how model parameters were estimated and model robustness assessed.

Address whether appropriate controls were implemented during your analyses and describe any replication measures or sensitivity analyses conducted to confirm the robustness and reliability of your results.

6. PLOS authors have the option to publish the peer review history of their article (what does this mean?). If published, this will include your full peer review and any attached files.

Reviewer #1: No

Reviewer #2: No

---

## [Author Response · Author response to Decision Letter 0]

13 Jun 2024

I have upload Response to Reviewers file.

---

## [Decision Letter · Decision Letter 1]

27 Aug 2024

The Impact of Green Technology Innovation on Carbon Emission Reduction Capacity in China: Based on Spatial Econometrics and Threshold Effect Analysis

PONE-D-23-42805R1

Dear Dr. Ma,

We’re pleased to inform you that your manuscript has been judged scientifically suitable for publication and will be formally accepted for publication once it meets all outstanding technical requirements.

Kind regards,

Pu-yan Nie

Academic Editor

PLOS ONE

Additional Editor Comments (optional):

Reviewers' comments:

Reviewer's Responses to Questions

**Comments to the Author**

1. If the authors have adequately addressed your comments raised in a previous round of review and you feel that this manuscript is now acceptable for publication, you may indicate that here to bypass the “Comments to the Author” section, enter your conflict of interest statement in the “Confidential to Editor” section, and submit your "Accept" recommendation.

Reviewer #1: All comments have been addressed

2. Is the manuscript technically sound, and do the data support the conclusions?

Reviewer #1: Yes

3. Has the statistical analysis been performed appropriately and rigorously? 

Reviewer #1: Yes

4. Have the authors made all data underlying the findings in their manuscript fully available?

Reviewer #1: Yes

5. Is the manuscript presented in an intelligible fashion and written in standard English?

Reviewer #1: Yes

6. Review Comments to the Author

Reviewer #1: The research meets all applicable standards for the ethics of experimentation and research integrity.

7. PLOS authors have the option to publish the peer review history of their article (what does this mean?). If published, this will include your full peer review and any attached files.

Reviewer #1: No

---

## [Editor Report · Acceptance letter]

1 Dec 2024

PONE-D-23-42805R1 

PLOS ONE

Dear Dr. Ma, 

I'm pleased to inform you that your manuscript has been deemed suitable for publication in PLOS ONE. Congratulations! Your manuscript is now being handed over to our production team.

Kind regards, 

on behalf of

Dr. Pu-yan Nie 

Academic Editor

PLOS ONE